# Reviews and syntheses: Trends in primary production in the Bay of Bengal – is it at a tipping point?

Carolin R. Löscher[1]

[1] Nordcee, DIAS, Department of Biology, University of Southern Denmark, Campusvej 55, 5230 Odense M- DK

*Correspondence to*: Carolin R. Löscher (cloescher@biology.sdu.dk)

**Abstract.** Ocean primary production is the basis of the marine food web, sustaining life in the ocean via photosynthesis, and
10 removing carbon dioxide from the atmosphere. Recently, a small but significant decrease of global marine primary production has been reported based on ocean color data, which was mostly ascribed to decreases in primary production in the northern Indian Ocean, particularly in the Bay of Bengal.

Available reports on primary production from the Bay of Bengal (BoB) are limited, and due to their spatial and temporal variability difficult to interpret. Primary production in the BoB has historically been described to be driven by diatom and
15 chlorophyte clades, while only more recent datasets also show an abundance of smaller, visually difficult to detect cyanobacterial primary producers. The different character of the available datasets, i.e. direct counts, metagenomic and biogeochemical data, and satellite-based ocean color observations, make it difficult to derive a consistent pattern. However, making use of the most highly resolved dataset based on satellite imaging a shift in community composition of primary producers is visible in the BoB over the last two decades. This shift is driven by a decrease in chlorophyte abundance, and a
20 coinciding increase in cyanobacterial abundance, despite stable concentrations of total chlorophyll. A similar but somewhat weaker trend is visible in the Arabian Sea, where satellite imaging points towards decreasing abundances of chlorophytes in the North and increasing abundances of cyanobacteria in the eastern parts. Statistical analysis indicated a correlation of this community change in the BoB to decreasing nitrate concentrations, which may provide an explanation for both, the decrease of eukaryotic nitrate-dependent primary producers and the increase of small unicellular cyanobacteria related to
25 *Prochlorococcus*, which have a comparably higher affinity to nitrate. Changes in community composition of primary producers and an overall decrease of system productivity would strongly impact oxygen concentrations of the BoB's low oxygen intermediate waters. Assuming decreasing nitrate concentrations and concurrent decreasing biomass production, export and respiration, oxygen concentrations within the oxygen minimum zone would not be expected to further decrease. This effect could be enhanced by stronger stratification as a result of future warming, and thus possibly counteract oxygen
decrease as a direct effect of stratification. Therefore, given a decrease in primary production, the BoB may not be at a tipping point for becoming anoxic, unless external nutrient inputs increase.

# 1 The role of the Bay of Bengal for primary production in the global Ocean – a historical perspective

Marine primary producers contribute around 50% to global net primary production (Behrenfeld et al. 2001), leading to a carbon flux from the atmosphere into the ocean of 45- 50 Pg C and up to 90 Pg C per year (Longhurst et al. 1995; Sabine et al. 2004; Sarmiento and Gruber 2002). Changes in ocean primary production exert an important control on atmospheric carbon dioxide ($CO_2$) concentrations, and thus on global climate (Falkowski, Barber, and Smetacek 1998). The BoB has often been described as an area of low primary production compared to the Arabian Sea. This low productivity has classically been ascribed to a strongly stratified water column as a result of increased surface water temperatures (Kumar et al. 2004) in combination with lowered surface water salinity due to monsoon-governed episodes of massive rainfall and river discharge with maximum freshwater inputs in September (e.g., Mahadevan (2016)). The stratification extends through large parts of the BoB basin (Subramanian 1993), restricting nutrient fluxes to the surface and eventually limiting primary production. In coastal areas, nutrient inputs from the major rivers have been described to stimulate primary production, however, rapid consumption as well as a ballasting effect with lithogenic particles and subsequent sedimentation of organic matter prevent offshore transport (Singh, Gandhi, and Ramesh 2012; Singh and Ramesh 2011; Krishna et al. 2016; Kumar et al. 2004; Ittekkot 1993). Open waters therefore appear low in macronutrients, exhibiting at least temporarily a slight nitrogen undersaturation (Bristow et al. 2017; Löscher et al. 2020). However, nitrogen fixation has been described low to non-existent (Saxena et al. 2020; Löscher et al. 2020), therefore not compensating the nitrogen deficit. The available geological record suggests that nitrogen fixation is generally absent since the last glacial maximum where isotope records showed and enrichment in [15]N indicative for nitrogen fixation (Contreras-Rosales et al. 2016; Shetye et al. 2014; Dähnke and Thamdrup 2013). Corresponding to this absence of $N_2$ fixation, low primary production is suggested from deep time records of total organic carbon (TOC, Fig. 1A) on a time scale of 18kyr before present (BP).

While a decrease in primary production has been derived in models for the last decades (Fig. 1B; (Gregg and Rousseaux 2019; Roxy et al. 2016)) in the Indian Ocean, shorter historical records of primary production in the BoB are not too abundant. However, records of direct rate measurements go back to the RV Galathea and RV Anton Bruun expeditions in the early 1950ies, followed by the International Indian Ocean Expedition (IIOE) from 1959 to 1965 (Snider 1961). Those earliest records report primary production of 0.1 – 2.16 mg C m$^{-2}$ d$^{-1}$ for the shelf regions and 0.1 - 0.3 mg C m$^{-2}$ d$^{-1}$ for open ocean waters of the BoB. Comparably higher rates were reported from an expedition with the Russian RV Vityaz from 1956 to 1960, with rates between 70 to 3600 mg C m$^{-2}$ d$^{-1}$, from a record from 1970 with a rate of 190 mg C m$^{-2}$ d$^{-1}$ (Nair et al. 1973), and from a summer monsoon situation in Aug./Sept. 1976 with rates between 130 and 330 mg C m$^{-2}$ d$^{-1}$ (Radhakrishna et al. 1978). Some of those earlier measurements were suggested to be biased as a result of trace metal contamination before trace metal clean techniques were available, a problem identified by calculating primary production to chlorophyll ratios, which turned out to be extremely high (250- 2500 compared to an average of $23 \pm 13$ in later data presented in Table 1; Madhupratap et al. (2003)). Later reports show a high variability of primary production ranging between 0.3 and 936 mg C m$^{-2}$ d$^{-1}$ (Gomes, Goes, and Saino 2000; Murty et al. 2000; Balachandran et al. 2008; Madhupratap

et al. 2003; Gauns et al. 2005; Kumar et al. 2010; Mohanty, Pramanik, and Dash 2014; Subha Anand et al. 2017; Löscher et al. 2020; Jyothibabu et al. 2004; Kumar et al. 2004; Madhu et al. 2006; Muraleedharan et al. 2007; Prasanna Kumar et al. 2002; Sarma et al. 2020; Saxena et al. 2020; Singh et al. 2015), and extremes of 2200 mg C m$^{-2}$ d$^{-1}$ (Bhattathiri, Devassy, and Radhakrishna 1980), with generally higher rates in shelf regions as compared to the open ocean, which were combined into average rates of 500 and 300 mg C m$^{-2}$ d$^{-1}$ for shelf and open ocean, respectively, to obtain a carbon flux budget (Naqvi et al.

2010). These average rates are quantitatively comparable to the studies presented in Table 1, however, for instance mesoscale water mass dynamics have been observed to promote primary production in the BoB beyond those ranges up to 920 mg C m$^{-2}$ d$^{-1}$, likely because of eddy-related decreases in stratification and pumping of nutrients into otherwise nutrient-exhausted photic surface waters (Sarma and Udaya Bhaskar 2018). Direct assessments of primary production in eddies of the BoB showed an increase in primary production and surface chlorophyll concentrations due to eddy-related nutrient pumping

(Singh et al. 2015; Sarma et al. 2020), with increased primary production being associated with diatom blooms (Vidya and Prasanna Kumar 2013). Eddies and other mesoscale and sub-mesoscale dynamics are frequent in the BoB (Cui, Yang, and Ma 2016; Greaser et al. 2020; Dandapat and Chakraborty 2016; Vimal Kumar et al. 2016), and therefore may cause significant variation in primary production patterns. Additional variation results from the strong influence of the two monsoon governed seasons on primary production ((Gomes, Goes, and Saino 2000; Jyothibabu et al. 2018; Madhu et al.

2002; Gauns et al. 2005); Table 1). Based on the presented data, a current estimate of primary production would be in the range of 361 ±145 and 236 ± 121 mg C m$^{-2}$ d$^{-1}$ for coastal and open ocean regions, respectively, which is one order of magnitude below the Arabian Sea, depending on the region and time of the year (Naqvi et al. 2010).

## 2 Key primary producers in BoB waters

Compared to records of primary production, even less data on the primary producer community are available, and

85 chlorophyll concentrations are often the only parameter presented (Table 1). Typically, coastal chlorophyll concentrations are about an order of magnitude higher compared to those in the central BoB (e.g. (Radhakrishna et al. 1978; Ramaiah et al. 2010; Balachandran et al. 2008; Gauns et al. 2005; Kumar et al. 2010). A detailed glider-based survey in the southern open ocean waters of the BoB recorded chlorophyll distributions with maxima of 0.3–1.2 mg m$^{-3}$ located at the base of the mixed layer at about 50 - 60 m water depth (Thushara et al. 2019). Records of discrete measurements show a comparable

distribution for the open waters of the BoB and in addition an extension of chlorophyll concentrations of up to 0.3 mg m$^{-3}$ north of 15˚N possibly connected to riverine nutrient imports (Bhushan et al. 2018; Löscher et al. 2020; Li et al. 2012). Exemplaric vertical profiles of open ocean chlorophyll distributions and a diversity of typically observable primary producers are depicted in Figure 2.

Historically available phytoplankton diversity records have methodological limitations relying mostly on direct or

95 microscopic phytoplankton counts; therefore, small sized phytoplankton and cyanobacteria are likely underrepresented. There is, however, a general consensus in earlier and newer studies that diatoms dominate the pool of primary producers

(Gauns et al. 2005; Madhupratap et al. 2003; Devassy, Bhattathiri, and Radhakrishna 1983), with some historical records being astonishingly detailed, presenting phytoplankton distribution down to the genus and species levels (Nair and Gopinathan 1983) and their results are comparable to more recent studies (Ramaiah et al. 2010) showing a diversity of diatoms including *Thalassiothrix, Nitzschia, Thalassionema, Skeletonema, Chaetoceros* and *Coscinodiscus* clades being abundant (Devassy, Bhattathiri, and Radhakrishna 1983; Ramaiah et al. 2010). Diversity analysis based on bulk DNA and amplicon sequencing complemented those previously available datasets by adding a higher diversity of eukaryotic phytoplankton, including *Pelagophyceae, Haptophyceae, Chrysophyceae, Eustigamatophyceae, Xanthophyceae, Cryptophyceae, Dictyochophyceae,* and *Pinguiophyceaeadding* and importantly by adding small cyanobacteria, which are difficult to count microscopically and were therefore not included into previous records (Löscher et al. 2020; Yuqiu et al. 2020; Bemal, Anil, and Amol 2019; Larkin et al. 2020; Pujari et al. 2019). Those cyanobacteria accounted for up to 60% of the primary producer abundance in sequence datasets in the central BoB (Li et al. 2012), and include *Synechococcus* and *Prochlorococcus*, the former has been detected from the surface down to the chlorophyll maximum, while the latter has been found abundant in the lower margin of the chlorophyll maximum at around 50 – 80 m water depth, slightly deeper than the maximum of eukaryotic primary producers (Löscher et al. 2020; Yuqiu et al. 2020). The *Prochlorococcus* population has been described to consist of several different ecotypes of the HLII clade with their respective abundances being governed by macro- and micronutrient distribution and by temperature (Larkin et al. 2020; Pujari et al. 2019). Similar distributions of *Prochlorococcus* and *Synechococcus* have been found in other OMZ areas (Beman and Carolan 2013; Franz et al. 2012; Meyer et al. 2015), following similar vertical and coast to open ocean patterns. The deeper maximum of *Prochlorococcus* as a result of its pigment composition adapted to lower light levels (Moore, Rocap, and Chisholm 1998; Rocap et al. 2003) possibly allows for utilization of nutrients from sinking organics matter at the lower boundary of the mixed layer. Metagenomes from the Atlantic have previously demonstrated the genetic potential of Prochlorococcus HLII clades to grow on nitrate (Rusch et al. 2007) supporting earlier suggestion that some *Prochlorococcus* ecotypes thrive at the base of the euphotic zone to acquire nitrate from underlying waters (Vaulot and Partensky 1992; Olson et al. 1990). While there is a body of literature describing distribution patterns of Prochlorococcus ecotype (e.g. (Johnson et al. 2006; Martiny, Kathuria, and Berube 2009; Moore, Rocap, and Chisholm 1998)), the relative contribution of different Prochlorococcus ecotypes to primary production in the ocean is not well resolved. In addition, information on the specific contribution of Prochlorococcus ecotypes detected in the BoB to bulk primary production is not available. Thus, it is unclear if a change in Prochlorococcus ecotype composition as suggested by Larkin et al., 2019, in response to changing temperatures, nutrient concentration, or iron stress would correspond to changes in overall Prochlorococcus primary production. A community shift in small cyanobacteria may be somewhat speculative and with unknown impacts on bulk primary production. However, an overall increase in abundance of small cyanobacteria in concert with a decrease of eukaryotic primary producers would be expected to impact BoB biogeochemistry, especially with regard to the spatial expansion and the intensity of the OMZ through modified export production and respiration in low oxygen intermediate waters.

Besides those small cyanobacteria, there are reports on nitrogen fixing cyanobacteria of the *Trichodesmium* clade (Devassy, Bhattathiri, and Radhakrishna 1983; Jyothibabu et al. 2017; Sahu et al. 2017; Hegde et al. 2008; Shetye et al. 2013; Wu et al. 2019); other reports included diatom-diazotroph associations playing a role for BoB nitrogen fixation (Bhaskar et al. 2007). However, for both types of nitrogen fixing primary producers, datasets are not conclusive and indicate high spatial and temporal variability. Nitrogen fixing microbes have been proposed to be limited by iron, other micronutrients or organic

matter in the BoB (Löscher et al. 2020; Saxena et al. 2020; Shetye et al. 2013; Benavides et al. 2018). While micronutrients would have the potential to also directly limit primary production, a limitation of nitrogen fixers by organic matter would result in a feedback regulation of low primary production limiting nitrogen fixation and resulting low nitrogen availability limiting primary production.

**3 Trends in primary production in the BoB**

    Satellite data from 1998 to 2015 suggest a decrease in primary production in the global ocean (Gregg et al. 2003; Behrenfeld et al. 2006), and recent studies deducted a decrease in ocean primary production of 2.1% per decade associated largely to a decrease of chlorophytes in the marine photic realm (Gregg, Rousseaux, and Franz 2017; Gregg and Rousseaux 2019). However, a recent study, derived a nonlinear trend in primary production from a similar time episode, between 1998 and

2018 (Kulk et al. 2020). Decreasing rates of primary production have been associated with high latitude regions (Gregg et al. 2003), but also with the Northern and Equatorial Indian Ocean with a decrease of 9.7 and 17.2 % per decade, respectively (Gregg and Rousseaux 2019). These estimates, based on satellite imaging, were explained by a decrease in diatom and chlorophyte primary production of 15.4 and 24.8 % per decade, respectively, for both the BoB and its sister basin, the Arabian Sea (Fig. 3). This decrease has been connected to decreasing nitrate and silicate concentrations of 32.4% and 22.8%

150     per decade in those waters limiting those larger, fast-growing primary producer groups (Gregg and Rousseaux 2019), with nitrate rather than silicate being limiting primary production if assuming Redfield stoichiometry (Kumar et al. 2010; Radhakrishna et al. 1978). At the same time, an increase in small cyanobacterial primary producers, *Prochlorococcus* and *Synechococcus*, was described in this region, with an increase in cyanobacterial primary production of 16.7 % per decade (Gregg and Rousseaux 2019). Satellite-based imaging indeed showed a southward expansion and increase in abundance of

cyanobacteria in the Bay of Bengal and through the Southern Arabian Sea (Fig. 3). Molecular genetic data showed, however, that not only *Prochlorococcus* is expanding but that mostly certain ecotypes of high light *Prochlorococcus* increased in abundance and extended their habitat (Larkin et al. 2020). Given the decrease in both nitrate and silicate, a decrease in the silicate correlated ecotypes currently dominant in the northern BoB would be expected and those may be replaced by an open ocean ecotype sensitive to increasing iron concentrations in those waters. The overall increase in cyanobacteria derived

from satellite monitoring is, however, not provable by direct measurements due to the lack of counts in the earlier reports, and further doesn't seem to impact the overall prediction on primary production decrease.

Qualitatively consistent with the short term trend of decreasing primary production between 1998 and 2015, a pronounced decrease of up to 20% in phytoplankton in the Western Indian Ocean over the past six decades has been ascribed to increasing ocean stratification as a consequence of rapid warming in the Indian Ocean, which suppresses nutrient mixing from subsurface layers (Roxy et al. 2016). This result is indeed consistent with a long term trend with decreasing productivity since the last glacial maximum (Contreras-Rosales et al. 2016; Shetye et al. 2014). With primary production leading to respiration and a concurrent oxygen loss in intermediate waters, this may provide an explanation for why the BoB is the only oxygen minimum zone region with traces of oxygen left in its core waters. It has often been suggested that the BoB is at a tipping point to developing severe anoxia (Bristow et al. 2017; Canfield et al. 2019; Rixen et al. 2020), which is a threshold with only minor changes in biogeochemistry leading to a consumption of oxygen traces in the oxygen minimum zone. This scenario is, however, challenged by decreasing primary production on long-term, as well as decadal time scales.

**4 Possible scenarios in response to changes in primary production on the BoB OMZ**

Reports of decreasing primary production in the BoB available from geological records, Earth system modelling, and satellite imaging are consistent over different time scales. But explanations on why primary production and chlorophyll concentrations decrease differ. Proposed important parameters include iron stress with iron concentrations having decreased in the geological record over the last 5000 years (Shetye et al. 2014), a decrease in nitrate and silicate availability directly impacting primary producer growth (Gregg and Rousseaux 2019), a rapid temperature increase of 0.6˚C over the last six decades, or a combination of those factors, which may directly or indirectly via increased stratification decrease primary production (Roxy et al. 2016). These considerations cannot clearly be compared and evaluated using the few direct measurements available, as those expose a high temporal and spatial variability. They allow, however, for exploring theoretically what would happen to the BoB biogeochemistry if nutrient concentrations would decrease further, with the exception of coastal regions, where nitrogen inputs may increase and enter the ocean via rivers but would also at the same time be removed quickly and close to the coast as currently happening in the BoB (Naqvi et al. 2010), and temperatures would increase.

Assuming a limitation of primary production by nitrogen availability, we would expect a niche for nitrogen fixation developing in the BoB. Until now, nitrogen fixation rates have shown to be low (Löscher et al. 2020; Saxena et al. 2020) and while there were reports on local blooms of the efficient nitrogen fixer *Trichodesmium* (Shetye et al. 2013), the nitrogen fixer community is dominated by typically less active heterotrophic bacteria (Wu et al. 2019; Turk-Kubo et al. 2014). However, our understanding of the diazotroph community composition and $N_2$ fixation rates is hampered by the low number of available datasets and their spatial and seasonal bias. Nitrogen fixers in general have a high requirement for iron, therefore an iron limitation could ultimately limit nitrogen fixation and indirectly primary production, as discussed earlier (Löscher et al. 2020). A further decrease in iron would intensify this limitation and progressive decrease the productivity in the BoB. In

addition, a decrease in silicate would limit diatom growth which need silica to form their frustules. Therefore, a combined decrease of iron, nitrate and silicate concentrations will lower primary production of various groups of primary producers at the same time, which may not only explain the trend visible from satellite imaging (Gregg and Rousseaux 2019) but may allow to predict a future trend for the BoB biogeochemistry.

Our earlier studies presented possible feedback cycles that are able to explain the persistent nanomolar levels of oxygen in the BoB OMZ (Canfield et al. 2019; Löscher et al. 2020). One approach included low mixing levels, or permanent stratification limiting euphotic zone nutrient concentrations to an extent that new production is persistently low and organic matter recycling will not support the organic carbon requirement of the detected heterotrophic nitrogen fixer community. This we suggested to lead the system to being locked in a low productivity and increasingly nitrogen limited scenario with the OMZ increasingly weakening. Applying lower concentrations of nitrate and iron (Fig. 4; silicate is not parametrized in our model but would lead to a similar effect if it would be limiting primary production), we observe that the OMZ respiration will lower, low oxygen concentrations will be maintained, and denitrification will only occur if nutrients are imported into the OMZ from land, via rivers, from the atmosphere, or by increased upwelling (Fig. 4). This would mean the BoB may not be at a tipping point towards anoxia but is a system with a weakening OMZ in its open waters, with progressive warming stabilizing this trend by increasing stratification in the photic zone and cutting this part of the water column off any nutrient supply.

Comparing scenarios of primary production and its impact on the BoB OMZ, the last glacial maximum signifies an episode of high productivity in the geological record (Contreras-Rosales et al. (2016), Fig. 5A). Higher land runoff and riverine inputs led to both higher nutrient imports, but also increased loads of terrigenous material leading facilitating organic carbon export from the productive zone to the sediments through ballasting. This effect is currently also seen, with nutrients being imported and consumed close to the shelf and organic material exported out of the photic zone (Fig. 5B), leading to a carbon pump with similar export rates like in the Arabian Sea (Singh and Ramesh 2015). A scenario with production being enhanced would strongly depend on external nutrient inputs, those could come from land and riverine inflow, and could for example result from deforestation, enhanced monsoon events, increasing atmospheric input, or enhanced upwelling, which has been described to enhance primary production in (sub-) mesoscale features (Sarma and Udaya Bhaskar (2018), Fig. 5B). Because global warming will result in increased stratification, enhanced nutrient pumping from deeper waters may be limited to mesoscale eddies, the BoB may be a rather stable system in itself and the observed and predicted changes in primary producers are not suggestive of a development of anoxia in the BoB OMZ.

Acknowledgement

I thank C. F. Reeder, P. Xu and J. Rønning, J. Lincy, and D. E. Canfield for helpful discussions on BoB productivity patterns and the BoB OMZ, I also thank the Villum foundation for funding my research (Grant no. 29411). I thank M. Benavides and A. Singh for their helpful and constructive reviews.

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

**Table 1: Historical record of water column integrated chlorophyll a concentration, surface chlorophyll a concentration, and primary production**

| Year | Month | Season | water column integrated Chl a [mg m$^{-2}$] | Surface Chl a [mg m$^{-3}$] | Primary production [mg C m$^{-2}$ d$^{-1}$] | Reference |
|---|---|---|---|---|---|---|
| 1951 | | | | | 0.1 - 2.16 | Galathea and Anton Brunn expedition, Steemann Nielsen and Jensen, 1957 |
| 1956 | | | | | 70 to 3600 | R.V. Vityaz in 1956-60 |
| 1961 | | | | | 190 | Nair 1970 |
| 1976 | | summer monsoon | 8.63 - 28.45 | 0.084 - 1.67 | 129.99 - 329.49 | Rhadakrishna et al 1978 |
| 1977 | | summer monsoon | 2.11 - 33.72 | 0.03 - 1.04 | | Devassy et al 1983 |
| 1978 | Aug | summer monsoon | 1.28 - 33.72 (up to 50) | 0.01 - 1.01 | 180 - 2200 | Bhattahiri et al. 1980 |
| 1996 | May/ June | summer monsoon | | 0.01 - 0.2 | | Murty et al 2000 |
| 1996 | April/ May | spring intermonsoon | up to 53 | | 4.5 | Gomes et al 2000 |
| 1996 | | summer monsoon | up to 92 | | 0.3 | Gomes et al 2000 |
| 2000 | July / Aug | summer monsoon | | | coastal: 350 ± 225 oceanic: 251 ± 177 | Madhu et al., 2006 |
| 2000 | Nov/ Dec | winter monsoon | | 9.0 - 15 | 87 - 187 | Balachandran et al 2008 |
| 2000 | Dec | winter monsoon | | | coastal: 252 ± 210 oceanic: 231 ± 150 | Madhu et al., 2006 |
| 2001 | Nov/ Dec | winter monsoon | coastal: 7 - 23 oceanic: 8 - 18 | coastal: 0.06 - 0.16 oceanic: 0.06 - 0.28 | coastal: 253 - 566 oceanic: 99- 423 | Gauns et al., 2005 Madhupratap et al. 2003 |
| 2001 | July/ Aug | summer monsoon | coastal: 12-19 oceanic: 10 – 11 | | coastal: 40 – 502 oceanic: 89 - 221 | Gauns et al. 2005 |
| 2002 | April | spring intermonsoon | | | coastal: 308 ± 120 oceanic: 303 ± 95 | Madhu et al., 2006 |
| 2002 | April/ May | spring intermonsoon | | 0.25 - 0.4 | | Prasanna Kumar et al., 2010 |
| 2002 | Sept/ Oct | intermonsoon | coastal: 11 - 19 oceanic: 13 - 16 | | coastal: 250 – 469 oceanic: 202 - 427 | Gauns et al., 2005 |
| 2002 | Nov/ Dec | winter monsoon | coastal: 9 – 15 oceanic: 9-13 | | coastal: 115 – 187 oceanic: 87 - 164 | Jyothibabu et al., 2004 |
| 2003 | April/ May | pre-monsoon | | | 154-975 (average coastal: 552, average oceanic: 284) | Kumar et al., 2004 |
| 2003 | July / Aug | summer monsoon | anticyclonic warm gyre: 1.84 cyclonic eddy: 5.01 upwelling zone: 5.2 | | anticyclonic warm gyre: negligible cyclonic eddy: 163 upwelling zone: 271 | Muraleedharan et al., 2006 |
| 2003 | Sept | summer monsoon | | 0.2 - 0.35 | 89.4 - 220.6 | Prasanna Kumar et al., 2010 |
| 2003 | | fall intermonsoon | | 0.3 - 0.4 | 184.14 - 512.85 | Prasanna Kumar et al., 2010 |
| 2003 | Sept/Oct | post-monsoon | | | coastal: 281 oceanic: 364 | Kumar et al., 2004 |
| 2007 | Nov/Dec | pre-/ early winter monsoon | | | cyclonic eddy: 203-430 | Singh et al., 2015 |
| 2010 | | summer | | | 221.41 ± 4.97 | Swati Sucharita Mohanty et al. 2014 |
| 2010 | | winter | | | 186.69 ± 9.87 | Swati Sucharita Mohanty et al. 2014 |
| 2010 | | monsoon | | | 151.25 ± 2.16 | Swati Sucharita Mohanty et al. 2014 |
| 2010 | | post-monsoon | | | 167.87 ± 3.02 | Swati Sucharita Mohanty et al. 2014 |
| 2014 | Jan | NE monsoon | | 0.08 - 0.035 | 1.4 - 9.3 | Löscher et al., 2020 |
| 2014 | March April | intermonsoon | | | 182 - 1261 (average 936 ± 350) | Anand et al., 2017 |
| 2018 | March/ April | intermonsoon | 34.6 ± 4 | cyclonic eddy 0.35 ± 0.08 | 411-920 | Sarma, et al., 2019 |
| 2018 | March/ April | intermonsoon | 26.4 ± 4 | outside eddy 0.22 ± 0.06 | | Sarma, et al., 2019 |
| 2018 | March/ April | intermonsoon | 23.6 ± 3 | anticyclonic eddy northern region 0.11 ± 0.06 | | Sarma, et al., 2019 |
| 2018 | March/ April | intermonsoon | 22.2 ± 3 | anticyclonic eddy southern region 0.10 ± 0.03 | | Sarma, et al., 2019 |
| 2018 | July/Aug | summer monsoon | | | 288-1044 | Saxena et al., 2020 |

**Figures**

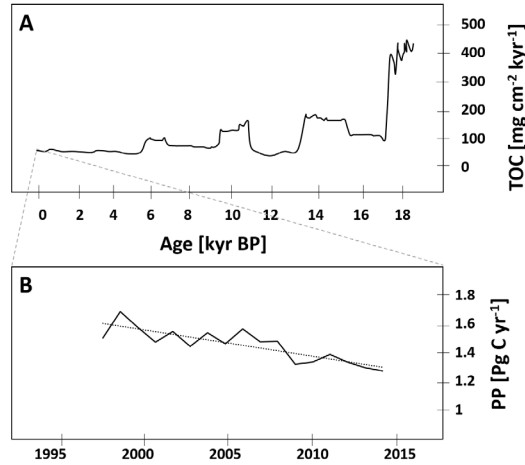

**Figure 1: (A) Trends or total organic carbon (TOC) in the sediment record over the last 18 ky before present adapted from Contreras-Rosales et al. (2016), and (B) modelled decrease in primary production between 1998 and 2014 modified from Gregg and Rousseaux (2019)**

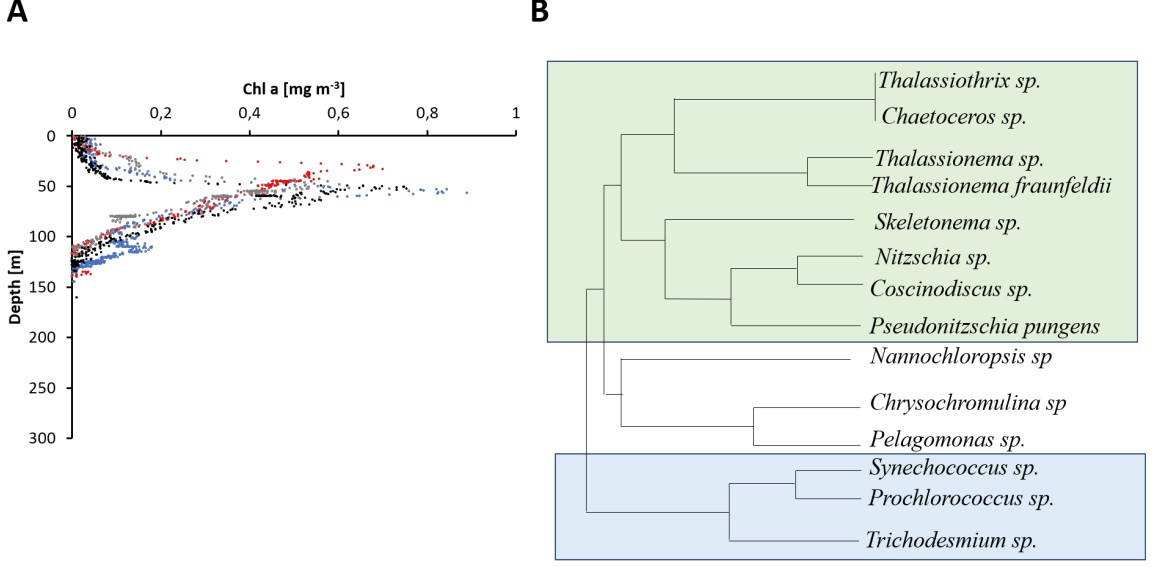

**Figure 2: (A) Vertical profiles of chlorophyll a from four stations in the open ocean region of the BoB taken from Löscher et al. (2020). (B) Schematic depiction of the phylogenetic diversity of primary producers identified in the BoB, green box: diatoms, blue box: cyanobacteria.**

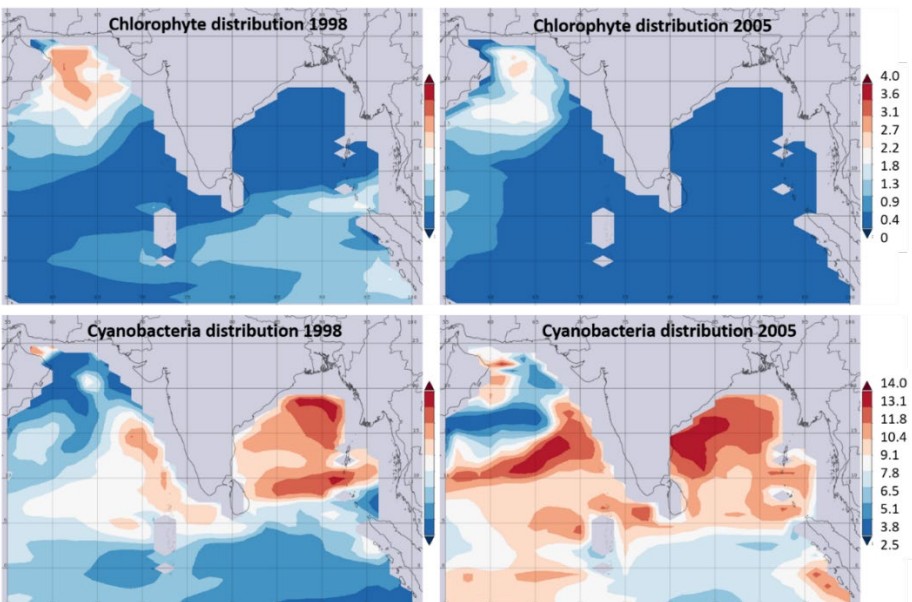

**Figure 3: Satellite imaging- based comparison of chlorophytes (top) and cyanobacteria from 1998 and 2015** in mg m⁻³ Data obtained from a combination of the Sea-viewing Wide Field of view Sensor (SeaWiFS), the Moderate Resolution Imaging Spectroradiometer (MODIS-Aqua), and the Visible Infrared Imaging Radiometer Suite (VIIRS) satellite product as available from https://giovanni.gsfc.nasa.gov have been averaged from Jan to 15Dec 1998 and 2005, respectively. The combination of those sensors allows for covering a range of different wavelengths useful to identify different phytoplankton clades. The maps have been generated using the NASA Ocean Biogeochemical Model (NOBM, (Gregg and Casey 2007)) using the most recent version of NASA ocean colour data product (R2014). NOBM is designed to represent open ocean areas (water depths > 200 m).

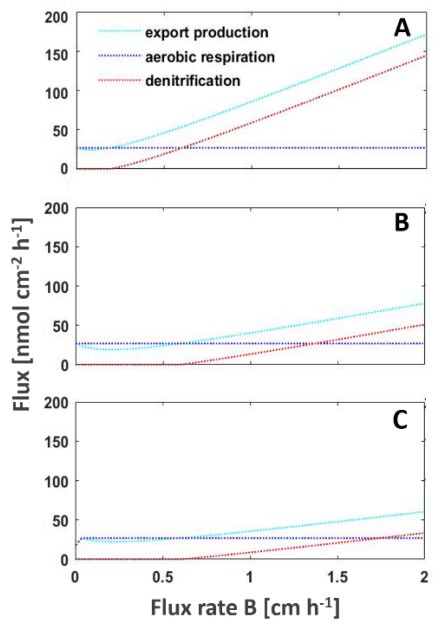

**Figure 4: Model of the main processes export production, aerobic respiration and denitrification shaping the intensity of the BoB OMZ to changing nutrient fluxes from either riverine, land, atmospheric inputs or upwelling to increased upwelling, with (A) current nutrient loads (B) decreased nitrate concentrations by 32% as predicted by Gregg et al. (2019) and (C) with both, decreased nitrate concentrations and decreased nutrient fluxes from deeper waters as a result of warming-dependent increased stratification. The model is adapted from Boyle et al. (2013).**

480

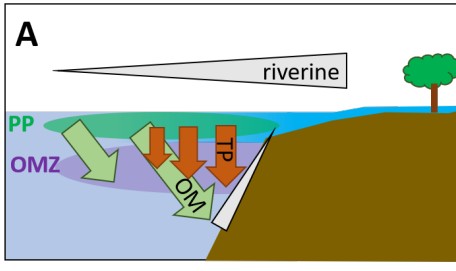

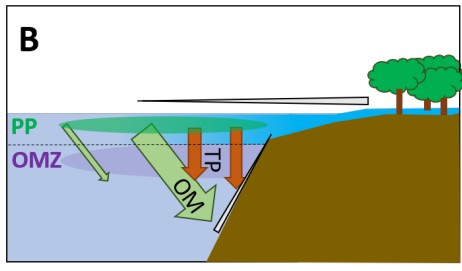

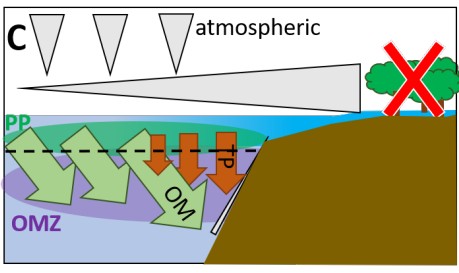

**Figure 5: Schematic depiction of fluxes impacting primary production and the oxygen minimum zone in the BoB (A) during the last glacial maximum (modified from Contreras-Rosales et al. (2016)), (B) currently, (C) and in a scenario leading to an anoxic OMZ.** The latter would require higher nutrient fluxes from either the atmosphere, from upwelling or from rivers and land. Fluxes from land may increase e.g. from deforestation, or enhanced rainfalls, however, higher terrigenous particle load would likely accompany increased nutrient loads and therefore, even if coastal primary production would increase, export production would increase via ballasting, too. The effect on the OMZ would then be rather small. Upwelling-dependent nutrient pumping is unlikely to increase due to warming and enhanced stratification but may occur especially in eddy systems, i.e. as eddy pumping. Grey triangles depict nutrient fluxes from land and rivers, from upwelling along the shelf and from the atmosphere, brown arrows depict terrigenous particle fluxes as imported from rivers and leading to ballasting and enhanced organic carbon export, green arrows depict organic material exported from the photic zone primary production (green bubble, PP) into the OMZ (purple bubble). Sizes of triangles, arrows, and bubbles qualitatively indicate proportions.