# Peer review of "Reviews and syntheses: Trends in primary production in the Bay of Bengal – is it at a tipping point?"

_Biogeosciences, 2020_

## Referee Comment (RC1)

Review manuscript bg-2020-494
*Reviews and syntheses: Trends in primary production in the Bay of Bengal – is it at a tipping point?*
by Carolin R. Löscher

The BoB has been hypothesized to be "at a tipping point" towards reaching severe anoxia in intermediate waters. Löscher provides an historical overview of primary productivity, nutrient limitation and phytoplankton community composition in the BoB. In the BoB, decreasing nutrient availability has caused a shift from large to small phytoplankton and a consequent decrease in primary productivity over the past two decades. Löscher challenges the view of the BoB as being about to become anoxic based on the apparent decrease in primary productivity, which would require less oxygen consumption in the water column.

The limited amount of in situ data available makes these interpretations considerably uncertain, but still of great use in providing a frame of discussion and inspiration for the upcoming activities of the IIOE-2.

Below I provide some minor comments and a few open questions to discuss with the author.

L29: Wouldn't stratification cause the opposite effect by impeding ventilation?

L49: Geological time scale is a bit vague; can you add the time span considered?

L50-51: empirical records?

L56: I would propose adding the time of year when the other rates were measured as a comparison to this indication of summer monsoon here.

L68 onwards: The role of mesoscale features is a bit downplayed, there have been a few publications on this for the BoB recently.

L88-89: better use "underrepresented"?

L107-108: Do you have a reference for this?

L112: Is it? do we have evidence that different community compositions of Synecho/Procholoro provide comparatively quantities of primary production?

L121-124: Organic matter availability would limit non-cyanobacterial diazotrophs, right? Unless you're considering a mixotrophic potential of the cyanobacterial ones (which I buy!).

L135 onwards: what is the cause of this tremendous decrease in nutrients?

L145: Can you elaborate, maybe earlier in the text when the different Prochloros are introduced, on their potential different contribution to primary production? Are some more productive than others? Or is their level of productivity spatiotemporally controlled by the availability/dynamics of the resources that are specifically limiting for each clade?

L149-150: The concept of the tipping point and the discussion around oxygen in the BoB needs a bit more detailed introduction for the non-familiarized reader.

L157 onwards: Probably all these factors co-occur and interact.

L168: Remove "could"?

L169: But point out the seasonal bias in measurements available so far.

L170: But there would probably be a sort of successional equilibrium where nutrient limitation would decrease primary production creating a niche for $N_2$ fixation, followed by DIN availability stemming from diazotrophs reviving non-diazotroph primary producers back. I guess the key question here is P availability.

L175: frustules?

L179-183: I agree that limited primary production due to stratification would lead to decreased export and less oxygen consumption in intermediate layers, but wouldn't it also impede ventilation?

L201: I would maybe refer to sub/mesoscale features in general, as features other than eddies (e.g. driven by wind) can drive localized vertical mixing.

---

## Referee Comment (RC2)

Carolin R. Löscher present synthesis of the primary production work in the Bay of Bengal. This article is quite comprehensive and utilized most of the published data. In addition, both the future simulations and geological records are presented, making this work quite unique and putting the Bay of Bengal in the context of the global ocean. It is concluded that the Bay may not be at a tipping point for becoming anoxic. My main comment is to improve the table 1 further with more studies from the Bay of Bengal as suggested below. Some minor comments are also listed:

Line 13: due their -> due to their

line 33: "carbon flux from the atmosphere into the ocean of 45- 50 Tg C per year". This flux is too low. This should be Peta-gram (Pg) per year, and even higher >90 Pg per year (Sabine et al., 2004; Sarmiento & Gruber, 2002). Longhurst et al. (1995) provided flux is Gt (giga tonnes) per year, which is same as Pg per year.

Line 47: delta -> δ

Line 47: "isotope records lastly were enriched for delta15N, indicative for nitrogen fixation". I understand what is meant here but it could mislead/misinterpreted as high $\delta^{15}N$ (enriched in 15N) would mean no nitrogen fixation.

Line 47: "enriched for delta 15N", there is a technical issue here. $\delta^{15}N$ is defined mathematically, it can be high/low, positive/negative but cannot be enriched/depleted. Reservoir may be enriched/depleted in $^{15}N$. So I would just say: "enriched in $^{15}N$"

Line 49: "on a geological time scale" is redundant as "geological record suggests" and "last glacial maximum" are parts of the same sentence.

Line 51: Indian ocean -> Indian Ocean

Line 54 and elsewhere: Primary production, by definition, is a rate of fixed carbon. So "rates" is redundant in "Primary production rates".

Lines 58-59: "Some of those earlier measurements were, however, likely biased as a result of trace metal contamination before trace metal clean techniques were available" Most (if not all) primary production incubations are not done in trace metal clean equipment.

Line 67: Table1 -> Table 1

Line 70: "three monsoon". winter and summer, what is the third one?

Line 73: "one to two orders of magnitude below the Arabian Sea" one sounds reasonable, two seems to be quite stretched.

Lines 111-115: seems some grammatical issue with the sentence.

Line 127: "Over the last two decades, primary production in the global ocean has decreased (Gregg et al. 2003; Behrenfeld et al. 2006)". Since the references are old, it will be helpful to mention the exact duration of the decade. Also, see this recent article by Kulk et al. (2020) that proposed a non-linear trend in productivity.

Line 179: cycles able -> cycles that are able

Table 1: Needs one more column that should include methodology of primary production measurements (such as $^{13}C$, $^{14}C$, $^{15}N$). There are some more studies that are missing in the list, such as: Jyothibabu et al., 2004; Kumar et al., 2004; Madhu et al., 2006; Muraleedharan et al., 2007; Prasanna Kumar et al., 2002; Sarma et al., 2020; Saxena et al., 2020; Singh et al., 2015.

Table 1: Madhupratap et al. 2001 should be replaced by Madhupratap et al. 2003.

Fig. 3 caption (line 403): Is it 1 Jan to 15 Dec or whole year 1 Jan – 31 Dec? Jan date should be mentioned.

Fig. 3: 1997-98 was one of the strongest el-nino years, so could this difference in the parameters for 1998 and 2005 be due to the natural climate variability (el-nino) and not the ongoing anthropogenic climate change?

References:

Jyothibabu, R., Maheswaran, P., Madhu, N., Ashraf, T. M., Gerson, V. J., Haridas, P., et al. (2004). Differential response of winter cooling on biological production in the northeastern Arabian Sea and northwestern Bay of Bengal. *Current Science*, 783–791.

Kulk, G., Platt, T., Dingle, J., Jackson, T., Jönsson, B. F., Bouman, H. A., et al. (2020). Primary production, an index of climate change in the ocean: satellite-based estimates over two decades. *Remote Sensing*, *12*(5), 826.

Kumar, S., Ramesh, R., Sardesai, S., & Sheshshayee, M. (2004). High new production in the Bay of Bengal: Possible causes and implications. *Geophysical Research Letters*, *31*(18).

Longhurst, A., Sathyendranath, S., Platt, T., & Caverhill, C. (1995). An estimate of global primary production in the ocean from satellite radiometer data. *Journal of Plankton Research*, *17*(6), 1245–1271.

Madhu, N., Jyothibabu, R., Maheswaran, P., Gerson, V. J., Gopalakrishnan, T., & Nair, K. (2006). Lack of seasonality in phytoplankton standing stock (chlorophyll a) and production in the western Bay of Bengal. *Continental Shelf Research*, *26*(16), 1868–1883.

Muraleedharan, K., Jasmine, P., Achuthankutty, C., Revichandran, C., Kumar, P. D., Anand, P., & Rejomon, G. (2007). Influence of basin-scale and mesoscale physical processes on biological productivity in the Bay of Bengal during the summer monsoon. *Progress in Oceanography*, *72*(4), 364–383.

Prasanna Kumar, S., Muraleedharan, P., Prasad, T., Gauns, M., Ramaiah, N., De Souza, S., et al. (2002). Why is the Bay of Bengal less productive during summer monsoon compared to the Arabian Sea? *Geophysical Research Letters*, *29*(24).

Sabine, C. L., Feely, R. A., Gruber, N., Key, R. M., Lee, K., Bullister, J. L., et al. (2004). The oceanic sink for anthropogenic $CO_2$. *Science*, *305*(5682), 367–371.

Sarma, V., Chopra, M., Rao, D., Priya, M., Rajula, G., Lakshmi, D., & Rao, V. (2020). Role of eddies on controlling total and size-fractionated primary production in the Bay of Bengal. *Continental Shelf Research*, *204*, 104186.

Sarmiento, J. L., & Gruber, N. (2002). Sinks for anthropogenic carbon. *Physics Today*, *55*(8), 30–36.

Saxena, H., Sahoo, D., Khan, M. A., Kumar, S., Sudheer, A., & Singh, A. (2020). Dinitrogen fixation rates in the Bay of Bengal during summer monsoon. *Environmental Research Communications*, *2*, 051007. https://doi.org/10.1088/2515-7620/ab89fa

Singh, A., Gandhi, N., Ramesh, R., & Prakash, S. (2015). Role of cyclonic eddy in enhancing primary and new production in the Bay of Bengal. *Journal of Sea Research*, *97*, 5–13.

---

## Author Response (AR1)

**Response to reviewer #1: Dr Mar Benavides**

Dear Dr Benavides

Thank you for the thorough review of my perspective paper, I am glad you found value in it and I thank you for your suggestions and for constructively discussing what is not clear.

I adressed your comments as follows:

L29: Wouldn't stratification cause the opposite effect by impeding ventilation?

➔ Yes, this would be the general assumption and is often the best line of reasoning. However, in a stagnant, undynamic system, a scenario with hampered ventilation will lead to an OMZ but the OMZ, in my view, can only lose all the oxygenif enough respiration takes place. This can only happen if enough nutrients are transported into the surface to allow for organic matter production, which then can be respired. Therefore, the question of how much oxygen is left would boil down to physical ventilation by water mass dynamics, organic matter respiration and topography. The latter would be important because if we have a shallower water column, we could assume that even lower organic matter export rates would lead to enough respiration to cause anoxia. In contrast, if we lose a lot of organic material to the deeper water column we would possibly not see the same effect. In the end it's a question of balance, or of what has the stronger effect. Because the BoB has this strongly stratified water column and ventilation would therefore not be a strong factor, my line of reasoning would be that the biological contribution to OMZ formation is the one to look at. I clarified my standpoint as follows: 'Assuming decreasing nitrate concentrations and concurrent decreasing biomass production, export and respiration, oxygen concentrations within the oxygen minimum zone would not be expected to further decrease. This effect could be enhanced by stronger stratification as a result of future warming, and thus possibly counteract oxygen decrease as a direct effect of stratification.'

L49: Geological time scale is a bit vague; can you add the time span considered?

➔ Agreed and changed to 'on a time scale of 18kyr before present (BP).'

L50-51: empirical records?

➔ That would be the problem, the available collection of empirical data as presented in table 1 is to scattered and patchy to derive a solid pattern from it, therefore, satellite based data are the only constant record available to us.

L56: I would propose adding the time of year when the other rates were measured as a comparison to this indication of summer monsoon here.

➔ I added the time frames to the summer monsoon situation (Aug.Sept 1976), the other two records do unfortunately not have an information on the monsoon situation.

L68 onwards: The role of mesoscale features is a bit downplayed, there have been a few publications on this for the BoB recently.

➔ I added a selection of recent studies:

Cui, Wei & Yang, Jungang & Ma, Yi. (2016). A statistical analysis of mesoscale eddies in the Bay of Bengal from 22–year altimetry data. Acta Oceanologica Sinica. 35. 16-27. 10.1007/s13131-016-0945-3.
Greaser, S. R., Subrahmanyam, B., Trott, C. B., & Roman-Stork, H. L. (2020). Interactions between mesoscale eddies and synoptic oscillations in the Bay of Bengal during the strong monsoon of 2019. Journal of Geophysical Research: Oceans, 125, e2020JC016772.

S. Dandapat and A. Chakraborty, "Mesoscale Eddies in the Western Bay of Bengal as Observed From Satellite Altimetry in 1993–2014: Statistical Characteristics, Variability and Three-Dimensional Properties," in IEEE Journal of Selected Topics in Applied Earth Observations and Remote Sensing, vol. 9, no. 11, pp. 5044-5054, Nov. 2016, doi: 10.1109/JSTARS.2016.2585179.

Vimal Kumar KG, Jayalakshmi KJ, Sajeev R, Gupta GVM (2016) Role of Mesoscale Eddies in the Distribution Pattern of Zooplankton Standing Stock of Western Bay of Bengal During Spring Transition. J Mar Biol Oceanogr 5:1. doi:10.4172/2324-8661.1000150

Arvind Singh, Naveen Gandhi, R. Ramesh, S. Prakash, Role of cyclonic eddy in enhancing primary and new production in the Bay of Bengal, Journal of Sea Research, Volume 97, 2015, Pages 5-13, ISSN 1385-1101,

Vidya, P. J., and Prasanna Kumar, S. (2013), Role of mesoscale eddies on the variability of biogenic flux in the northern and central Bay of Bengal, J. Geophys. Res. Oceans, 118, 5760– 5771, doi:10.1002/jgrc.20423.

L88-89: better use "underrepresented"?

➔ Changed accordingly.

L107-108: Do you have a reference for this?

➔ Yes, this is indeed mentioned in e.g. Moore, Rocap, and Chisholm 1998, but also in Rocap et al., 2003, which has been added.

Rocap G, Larimer FW, Lamerdin J, Malfatti S, Chain P, Ahlgren NA, Arellano A, Coleman M, Hauser L, Hess WR, Johnson ZI, Land M, Lindell D, Post AF, Regala W, Shah M, Shaw SL, Steglich C, Sullivan MB, Ting CS, Tolonen A, Webb EA, Zinser ER, Chisholm SW. Genome divergence in two

Prochlorococcus ecotypes reflects oceanic niche differentiation. Nature. 2003 Aug 28;424(6952):1042-7. doi: 10.1038/nature01947. Epub 2003 Aug 13. PMID: 12917642.

L112: Is it? do we have evidence that different community compositions of Synecho/Procholoro provide comparatively quantities of primary production?

➔ Actually you may have a point, here, we do not really know in the case of those specific clades of Synechococcus/ Prochlorococcus. We do have an idea of the overall difference between Prochlorococcus, Synechococcus and picoeukaryotes, described e.g. in Liang et al 2017 and Buitenhus et al, 2012, to be around 36, 255, and 2590 fg C $cell^{-1}$ for Prochlorococcus, Synechococcus, and picoeukaryotes.
The different clades occupy different niches, they are different in several regards and they may produce different rates, this would make sense but we still may not have a good enough grasp of how the clades in the BoB contribute to primary production. I therefore changed the sentence to 'A community shift in small cyanobacteria may be somewhat speculative and with unknown impacts on bulk primary production. However, an overall increase in abundance of small cyanobacteria in concert with a decrease of eukaryotic primary producers would be expected to impact BoB biogeochemistry, especially with regard to the spatial expansion and the intensity of the OMZ through modified export production and respiration in low oxygen intermediate waters.'

Buitenhuis, E. T., Li, W. K. W., Vaulot, D., Lomas, M. W., Landry, M. R., Partensky, F., et al. (2012). Picophytoplankton biomass distribution in the global ocean. *Earth Syst. Sci. Data* 4, 37–46. doi: 10.5194/essd-4-37-2012

Liang Y, Zhang Y, Wang N, Luo T, Zhang Y and Rivkin RB (2017) Estimating Primary Production of Picophytoplankton Using the Carbon-Based Ocean Productivity Model: A Preliminary Study. Front. Microbiol. 8:1926. doi: 10.3389/fmicb.2017.01926

L121-124: Organic matter availability would limit non-cyanobacterial diazotrophs, right? Unless you're considering a mixotrophic potential of the cyanobacterial ones (which I buy!).

➔ Both, as we know from the cited studies, and in addition from your study in Frontiers, 2018, I added the latter one to make the case clearer:

Benavides Mar, Martias Chloé, Elifantz Hila, Berman-Frank Ilana, Dupouy Cécile, Bonnet Sophie,Dissolved Organic Matter Influences $N_2$ Fixation in the New Caledonian Lagoon (Western Tropical South Pacific), Frontiers in Marine Science, Vol 5, 2018, DOI:10.3389/fmars.2018.00089

L135 onwards: what is the cause of this tremendous decrease in nutrients?

It is associated with a shallowing of the surface mixed layer. This shallowing is suggested to reduce the supply of nutrients to the surface from deeper waters.

L145: Can you elaborate, maybe earlier in the text when the different Prochloros are introduced, on their potential different contribution to primary production? Are some more productive than others? Or is their level of productivity spatiotemporally controlled by the availability/dynamics of the resources that are specifically limiting for each clade?

This is an interesting topic and issue you are raising, here. Most studies focus either on the distribution of the various ecotypes, other studies talk about bulk rates produced by Prochlorococcus. The authors of the cited study (Larkin et al) conclude that the ecotypes will distribute in response to biogeochemical factors, rates are, however, not metioned. For the specific ecotypes in the BoB it will be difficult to say if there is a difference in primary production per ecotype or if the productivity will be the same in general but vary in response to biogeochemical variables. For other ocean regions and for cultures, e.g. Moore et al, 1998, described the dependency of primary production on different levels of light in Prochlorococcus ecotypes. A more deteailed assessment of the ecotype distribution patterns is available, e.g in Johnson et al, 2006, where the authors found Prochlorococcus ecotypes reflecting biogeochemical regimes. Martiny et al, 2009 describe a distribution of different ecotypes depending on the nitrogen source and their potential to take up nitrate specifically. I included those references and a description of this issue not being resolved in l. 111:

'While there is a body of literature describing distribution patterns of Prochlorococcus ecotype (e.g. Johnson et al., 2006, Martiny et al, 2009, Moore et al., 1998), the relative contribution of different Prochlorococcus ecotypes to primary production in the ocean is not well resolved. In addition, information on the specific contribution of Prochlorococcus ecotypes detected in the BoB to bulk primary production is not available. Thus, it is unclear if a change in Prochlorococcus ecotype composition as suggested by Larkin et al., 2019, in response to changing temperatures, nutrient concentration, or iron stress would correspond to changes in overall Prochlorococcus primary production rates.'

Zackary I. Johnson, Erik R. Zinser, Allison Coe,Nathan P. McNulty, Malcolm S. Woodward, Sallie W. Chisholm, 2006, Niche Partitioning Among Prochlorococcus Ecotypes Along Ocean-Scale Environmental Gradients, Science.

Martiny AC, Kathuria S, Berube PM. Widespread metabolic potential for nitrite and nitrate assimilation among Prochlorococcus ecotypes. Proc Natl Acad Sci U S A. 2009;106(26):10787-10792. doi:10.1073/pnas.0902532106

Moore, L., Rocap, G. Chisholm S., 1998, Physiology and molecular phylogeny of coexisting Prochlorococcus ecotypes, Nature, Vol 393

L149-150: The concept of the tipping point and the discussion around oxygen in the BoB needs a bit more detailed introduction for the non-familiarized reader.

I split the sentence and added the following:

'This result is indeed consistent with a long term trend with decreasing productivity since the last glacial maximum (Contreras-Rosales et al. 2016; Shetye et al. 2014).  With primary production leading to respiration and a concurrent oxygen loss in intermediate waters, this may provide an explanation for why the BoB is the only oxygen minimum zone region with traces of oxygen left in its core waters. It has often

been suggested that the BoB is at a tipping point to developing severe anoxia (Bristow et al. 2017; Canfield et al. 2019; Rixen et al. 2020), which is a threshold with only minor changes in biogeochemistry leading to a consumption of oxygen traces in the oxygen minimum zone. This scenario is, however, challenged by decreasing primary production on long-term, as well as decadal time scales.'

L157 onwards: Probably all these factors co-occur and interact.

➔ True, I added ' or a combination of those factors' to l. 162.

L168: Remove "could"?

➔ It's been removed

L169: But point out the seasonal bias in measurements available so far.

➔ I added the following statement in l. 171: However, our understanding of the diazotroph community composition and N2 fixation rates is hampered by the low number of available datasets and their spatial and seasonal bias.

L170: But there would probably be a sort of successional equilibrium where nutrient limitation would decrease primary production creating a niche for N2 fixation, followed by DIN availability stemming from diazotrophs reviving non-diazotroph primary producers back. I guess the key question here is P availability.

➔ This is what we would classically expect but it doesn't happen, maybe it doesn't happen just yet.

L175: frustules?

➔ Yes, changed accordingly.

L179-183: I agree that limited primary production due to stratification would lead to decreased export and less oxygen consumption in intermediate layers, but wouldn't it also impede ventilation?

➔ Agreed, however, as already discussed above, this is a question of oxygen import versus respiration. If we assume hampered ventialtion beyond what we have now, it would take less respiration to cause anoxia, but we would still need some respiration. Therefore it would boil down to which process responds faster, i.e. if we lose oxygen via stratification and lower ventilation or if we safe oxygen because less is respired because of lower nutrient fluxes due to stratification. What we see now are traces of oxygen maintained in the OMZ with an extreme stratification already, the enigma we don't understand making it cruicial to go back to the BoB and do organic matter addition experiments as well as oxygen additions in OMZ-water incubations.

L201: I would maybe refer to sub/mesoscale features in general, as features other than eddies (e.g. driven by wind) can drive localized vertical mixing

➔ Correct, and changed accordingly. I, admittedly, have a weakness for eddies.

Again, thank you for your time to review this manuscript, your ideas, questions and suggestions were very helpful and appreciated.

All the best

Carolin Löscher

**Response to reviewer #2 Dr Arvind Singh**

Dear Dr Singh,
Thank you for this encouraging and constructive review, I am especially grateful for providing missing information and references for table 1. It is indeed quite a challenge to find everything and I am happy that you kindly provided what I didn't see.
In the revised version of the manuscript I included missing references/ information in table 1 as suggested, and included all your other comments:

Line 13: due their -> due to their

➔ changed accordingly

line 33: "carbon flux from the atmosphere into the ocean of 45- 50 Tg C per year". This flux is too low. This should be Peta-gram (Pg) per year, and even higher >90 Pg per year (Sabine et al., 2004; Sarmiento & Gruber, 2002). Longhurst et al. (1995) provided flux is Gt (giga tonnes) per year, which is same as Pg per year.

➔ Apologies for the confusion, which has been modified to 'carbon flux from the atmosphere into the ocean of 45- 50 Pg C and up to 90 Pg C per year (Sabine et al., 2004; Sarmiento & Gruber, 2002; Longhurst et al., 1995).'

Line 47: delta -> δ

➔ changed accordingly

Line 47: "isotope records lastly were enriched for delta15N, indicative for nitrogen fixation". I understand what is meant here but it could mislead/misinterpreted as high δ15N (enriched in 15N) would mean no nitrogen fixation.

➔ Good point, this resulted from a somewhat narrow nitrogen fixation perspective, I have changed it to 'where isotope records showed and enrichment in $^{15}$N indicative for nitrogen fixation'.

Line 47: "enriched for delta 15N", there is a technical issue here. $\delta15N$ is defined mathematically, it can be high/low, positive/negative but cannot be enriched/depleted. Reservoir may be enriched/depleted in 15N. So I would just say: "enriched in $^{15}N$"

➔ Agreed, and changed accordingly, this is actually an important distinction.

Line 49: "on a geological time scale" is redundant as "geological record suggests" and "last glacial maximum" are parts of the same sentence.

➔ It is redundant and has been removed.

Line 51: Indian ocean -> Indian Ocean

➔ Changed accordingly.

Line 54 and elsewhere: Primary production, by definition, is a rate of fixed carbon. So "rates" is redundant in "Primary production rates".

➔ Changed throughout the manuscript.

Lines 58-59: "Some of those earlier measurements were, however, likely biased as a result of trace metal contamination before trace metal clean techniques were available" Most (if not all) primary production incubations are not done in trace metal clean equipment.

➔ I was not aware of this, I was thinking along the lines of acid washing etc and cited another collection of data, which essentially referred to rates having possibly be boosted by unwanted trace metal additions. However, this is actually a valid point, how would we know. The rates are probably as good as they are now. I kept the statement as it is a reference to an earlier assessment but weakened it 'Some of those earlier measurements were suggested to be biased as a result of trace metal contamination before trace metal clean techniques were available, a problem identified by calculating primary production to chlorophyll ratios, which turned out to be extremely high …'

Line 67: Table1 -> Table 1

➔ Changed accordingly.

Line 70: "three monsoon". winter and summer, what is the third one?

➔ Changed to 'two'.

Line 73: "one to two orders of magnitude below the Arabian Sea" one sounds reasonable, two seems to be quite stretched.

➔ The statement is based on the overviews provided by Naqvi and colleagues from 2010, but I agree, if we compare the data conservatively, one order of magnitude is correct. Therefore, this has been changed.

Lines 111-115: seems some grammatical issue with the sentence.

➔ Agreed, see also the response to Dr Benavides's comments, the sentence has been changed to: 'A community shift in small cyanobacteria may be somewhat speculative and with unknown impacts on bulk primary production. However, an overall increase in abundance of small cyanobacteria in concert with a decrease of eukaryotic primary producers would be expected to impact BoB biogeochemistry, especially with regard to the spatial expansion and the intensity of the OMZ through modified export production and respiration in low oxygen intermediate waters.'

Line 127: "Over the last two decades, primary production in the global ocean has decreased (Gregg et al. 2003; Behrenfeld et al. 2006)". Since the references are old, it will be helpful to mention the exact duration of the decade. Also, see this recent article by Kulk et al. (2020) that proposed a non-linear trend in productivity.

➔ I added the time frame and modified the sentence to the following: 'Satellite data from 1998 to 2015 suggest a decrease in primary production in the global ocean (Gregg and Rousseaux 2019), and recent studies deducted a decrease in ocean primary production of 2.1% per decade associated largely to a decrease of chlorophytes in the marine photic realm (Gregg, Rousseaux, and Franz 2017; Gregg and Rousseaux 2019). However, a recent study, derived a nonlinear trend in primary production from a similar time episode, between 1998 and 2018 (Kulk et al. 2020).'

Line 179: cycles able -> cycles that are able

➔ Changed accordingly.

The suggested references were included in the manuscript.

Thanks, again for the helpful insights.

All the best

Carolin Löscher

**Note to the editor:**

Dear Dr Menezes

I took the freedom to thank both reviewers for their constructive comments in the acknowledgements.

All the best

Carolin Löscher

Gregg, W. W., C. S. Rousseaux, and B. A. Franz. 2017. 'Global trends in ocean phytoplankton: a new assessment using revised ocean colour data', *Remote Sens Lett*, 8: 1102-11.

Gregg, Watson W., and Cecile S. Rousseaux. 2019. 'Global ocean primary production trends in the modern ocean color satellite record (1998–2015)', *Environmental Research Letters*, 14: 124011.

Kulk, Gemma, Trevor Platt, James Dingle, Thomas Jackson, Bror F. Jönsson, Heather A. Bouman, Marcel Babin, Robert J. W. Brewin, Martina Doblin, Marta Estrada, Francisco G. Figueiras, Ken Furuya, Natalia González-Benítez, Hafsteinn G. Gudfinnsson, Kristinn Gudmundsson, Bangqin Huang, Tomonori Isada, Žarko Kovač, Vivian A. Lutz, Emilio Marañón, Mini Raman, Katherine Richardson, Patrick D. Rozema, Willem H. van de Poll, Valeria Segura, Gavin H. Tilstone, Julia Uitz, Virginie van Dongen-Vogels, Takashi Yoshikawa, and Shubha Sathyendranath. 2020. 'Primary Production, an Index of Climate Change in the Ocean: Satellite-Based Estimates over Two Decades', *Remote Sensing*, 12: 826.